# Individual and Simultaneous Photodegradation of Trimethoprim and Sulfamethoxazole Assessed with the Microbial Assay for Risk Assessment

**DOI:** 10.3390/molecules30091907

**Published:** 2025-04-25

**Authors:** Grzegorz Nałęcz-Jawecki, Milena Wawryniuk, Weronika Kopczyńska, Joanna Giebułtowicz

**Affiliations:** 1Department of Toxicology and Food Science, Medical University of Warsaw, 02-091 Warszawa, Poland; 2Department of Drug Chemistry, Pharmaceutical and Biomedical Analysis, Medical University of Warsaw, 02-091 Warszawa, Poland; jgiebultowicz@wum.edu.pl

**Keywords:** MARA, photodegradation, sulfonamides, sunlight simulator, pharmaceuticals, antibiotics

## Abstract

Co-trimoxazole is an antibacterial drug, a mixture of sulfamethoxazole (SMX) and trimethoprim (TMP) in a ratio of 5:1. Due to the different susceptibility of both components to decomposition under the influence of sunlight, the aim of the study was to assess the change in bacteriostatic activity during irradiation of the mixture of these antibiotics in a sunlight simulator. The bacteriostatic activity was assessed using a Microbial Assay for Risk Assessment (MARA), monitoring drug concentrations as well as the formation of photodegradation products using liquid chromatography (LC). The toxicity analysis of the SMX and TMP mixture showed synergistic bacteriostatic activity for six bacterial strains. This activity was maintained even during sample irradiation when 80–90% of SMX was degraded. This may indicate the bacteriostatic effect of SMX photoproducts and/or the lack of necessity to maintain a 5:1 ratio between SMX and TMP to maintain a strong effect of the mixture of these antibiotics. Analysis using LC with a high-resolution mass spectrometry detector revealed the presence of 11 SMX degradation products, including two with preserved sulfonamide structure, which may exhibit bacteriostatic activity.

## 1. Introduction

Antibiotics are one of the most important groups of pharmaceuticals and personal care products (PPCPs) present in the environment due to their strong effect on microorganisms, both pathogenic and those necessary for the proper functioning of aquatic ecosystems [1]. Additionally, their presence in sublethal concentrations may lead to the development and promotion of microbial resistance to antibiotics [2,3]. Sulfonamides are the oldest group of antibiotics, introduced in the mid-20th century [4]. Currently, they are used in both human and veterinary medicine, constituting one of the most commonly used groups of antibiotics [5]. Among veterinary drugs, sulfonamides rank 3rd, with total sales of 417.3 tons of active substance per year in 31 European countries [6]. Their mechanism of action is based on a structural analogy with p-aminobenzoic acid, which is necessary in most bacteria for the synthesis of folic acid [7]. Due to the development of resistance to this action of many bacteria, medicine often uses a combination of sulfonamides with drugs with other mechanisms of action. Co-trimoxazole is a combination of sulfamethoxazole (SMX) and trimethoprim (TMP) in a 5:1 ratio. TMP, like sulfonamides, inhibits the synthesis of bacterial folic acid but at a different stage. TMP is a dihydrofolate reductase inhibitor that blocks the conversion of dihydrofolate to tetrahydrofolic acid [8]. Hence, co-trimoxazole has a much broader spectrum of bacteriostatic action and is less susceptible to the development of bacterial resistance.

SMX has been detected widely in rivers worldwide, reaching concentrations in the µg L^−1^ range, e.g., up to 1.77 and 4.9 µg L^−1^ in Poland and China, respectively [9,10,11]. TMP was detected in µg L^−1^ concentrations in rivers and wastewater treatment plant effluents [8,11]. The monitoring of both antibiotics was recommended in the inland waters of the European Union by placing them on both the 3rd [12] and 4th watch lists [13]. Like other PPCPs, they could undergo both biotic and abiotic changes in the environment, primarily photodegradation [8,14,15,16,17]. Many of the SMX and TMP transformation products have been well characterized chemically [14,15,18,19,20,21]. However, their effects on living organisms have been poorly studied. The most commonly used bioassay to assess the biological activity of the photodegradation process is the short-term test using the luminescent bacteria *Aliivibrio fischeri* [8]. It allows only the detection of acute effects of toxic substances; therefore, antibiotics and their transformation products were mostly non-toxic in this test, both in the case of SMX and TMP [7,8,22]. It does not allow for the recognition of changes in the bacteriostatic activity of samples during the process. Majewsky et al. [7], comparing the short- and long-term effects of SMX on *Vibrio fischeri* bacteria luminescence, showed that only the 24 h test is suitable for analysis of the bacteriostatic effect. They found that this effect of the transformation products was comparable to that of the parent compound. However, in the case of TMP, the inhibition of *Escherichia coli* bacterial growth was proportional to the antibiotic concentration, regardless of the formed oxidation products [23]. Thus, the use of only one bacterial strain in toxicity studies on antibiotics may be insufficient due to the widespread resistance of bacteria. *E. coli* ATCC 25922 strain was sensitive to TMP and resistant to SMX, and the 24 h EC_50_ values were 0.31 and 17.1 mg L^−1^, respectively [22].

Microbial Assay for Risk Assessment (MARA) is a multi-species biotest that enables the assessment of chronic toxicity on microorganisms belonging to different phylogenetic classes, including Gram-positive, Gram-negative, and one strain of fungi [24]. The unique result obtained in this test is a “toxic fingerprint”, which allows the assessment of the relative sensitivity of each of the 11 strains of microorganisms used [25]. Based on the fingerprint, it is possible to compare the bacteriostatic activity profiles of different samples, for example, antibiotics [26], as well as to learn about changes in the sample profile, e.g., under the influence of photodegradation.

Due to the widespread use of co-trimoxazole, SMX and TMP may occur together in the aquatic environment. It is interesting to know how the photodegradation of the mixture of both antibiotics proceeds and whether the effect of SMX photoproducts is also additive with TMP. Additionally, SMX and TMP photodegrade at different rates [18,20]. This raises the question of whether their additive effect requires maintaining a 5:1 ratio of the mixture ingredients.

The aim of this study was to comprehensively evaluate the photodegradation of SMX and TMP tested individually and in a 5:1 mixture using a SunTest CPS+ sunlight simulator. To evaluate the effect of formulation on this process, both a pure antibiotic mixture and an extract of Biseptol pills were tested. For the first time, a multi-species bacterial test was used in the analysis of photodegradation of antibiotic mixtures, enabling the assessment of the bacteriostatic activity of the entire sample against different strains of microorganisms. The three research hypotheses considered in the study were that (i) SMX and TMP photodegradation products exhibit residual bacteriostatic properties, (ii) the spectrum of bacteriostatic action of photoproducts is the same as that of the parent compounds, and (iii) SMX and TMP act additively with the degradation products of the other drug.

## 2. Results

The SMX and TMP solutions, their mixtures (5:1), and Biseptol extract were irradiated in SunTest CPS+ sunlight simulator for 30 and 60 min. The antibiotics concentration was assessed using HPLC with a diode array detector (HPLC–DAD), while the photoproducts were identified using LC with a high-resolution mass spectrometer (HR MS/MS). The bacteriostatic activity of the unirradiated and irradiated samples was assessed using the MARA assay. Finally, the toxicity of the samples was compared with the toxicity predicted based on the antibiotic concentrations in the solution.

### 2.1. Analysis of Photodegradation with HPLC-DAD

The SunTest CPS+ sunlight simulator emits radiation in the range of 290–800 nm (Figure 1a). The absorption spectra of SMX (Figure 1b) and TMP (Figure 1d) indicate that SMX should be directly photodegraded to a much greater extent than TMP. The HPLC–DAD analysis indicates that after just 30 min of irradiation, the concentration of SMX decreased 4-fold, regardless of whether the compound was irradiated individually, in a mixture with TMP, or in a Biseptol extract (Figure 2a). The next 30 min of irradiation caused a further, more than two-fold decrease in SMX concentration. However, the TMP concentration decreased only by 6–19% during 60 min of irradiation, with the highest decrease noted in the mixture of standard TMP and SMX solutions (Figure 2b).

### 2.2. Toxicity of SMX and TMP in the MARA

The MARA test uses 11 strains of microorganisms belonging to different phylogenetic groups. Table 1 shows the microbics toxic concentration (MTC) values of both antibiotics tested. SMX inhibited the growth of six strains of microorganisms, mainly Gram-positive bacteria. The most sensitive was *Brevundimonas diminuta*, followed by *Delftia acidovorans*, with MTC values of 0.17 and 1.34 mg L^−1^, respectively. In contrast, TMP, in the tested concentration range up to 20 mg L^−1^, inhibited the growth of 10 of 11 strains. The only exception was *B. diminuta*. The most sensitive strain was *Delftia acidovorans* with an MTC value of 0.054 mg L^−1^, and in as many as six cases, the MTC values were lower than 1 mg L^−1^ (Table 1).

The fingerprints presented in Figure 3 illustrate the relative toxicity of the tested antibiotics towards all test strains. In the case of drugs tested individually, a large advantage of the sensitivity of the most sensitive strains, number 2 (SMX) and 6 (TMP), is visible (Figure 3a,d). However, in the case of a mixture of antibiotics and Biseptol extract, a synergistic effect occurred and the growth of all test strains was inhibited (Figure 3g,j).

The fingerprint changed significantly during SMX exposure in the SunTest CPS+ apparatus (Figure 3b,c). A significant reduction in toxicity towards strain no. 2 was accompanied by a relative increase in toxicity towards strain nos. 3 and 6. Interestingly, the irradiated SMX sample inhibited the growth of *Pichia anomala* yeast. As expected, due to only minor changes in TPM concentration, the fingerprint did not change during irradiation of this antibiotic (Figure 3e,f). In the fingerprint image of the Biseptol extract and the mixture of SMX and TMP, a gradual decrease in relative toxicity towards strain no. 2 can be observed (Figure 3h,i,k,l). This is understandable due to the significant decrease in the concentration of SMX responsible for the toxic effect towards *B. diminuta*. In the case of the remaining strains, the growth of which was inhibited by both antibiotics, irradiation did not cause significant changes in relative toxicity. Fingerprints of mixture samples irradiated for 30 and 60 min were similar to those of TMP.

### 2.3. Predicted and Measured Toxicity Units

Using Equation (2) (Section 4.5), based on the MTC values and the actual concentrations of SMX and TMP, the predicted toxicity unit values (PTU) were calculated for each tested microbial strain. They were then compared with the measured toxicity unit values (MTU) calculated from Equation (3) (Section 4.5). The results are presented in Figure 4 and Figure 5.

In the case of individually irradiated SMX and TMP, the measured toxicity did not differ significantly from the predicted toxicity. This means that the resulting photodegradation products did not have a bacteriostatic effect on the tested strains. The non-irradiated mixture of SMX and TMP and the extract of Biseptol acted synergistically on 6 of 10 MARA bacterial strains (Figure 4). The MTU values were from 2.0 to 5.7 times higher than their corresponding PTU values. The largest differences, over 5-fold, were observed for *Microbacterium* sp. (Figure 4a). This effect was noted not only for strains sensitive to SMX at concentrations below 20 mg L^−1^ but also for those whose MTC values were above the highest concentration tested *Microbacterium* sp., *E. casseliflavus*, and *S. rubidea* (Figure 4a,d,f). This means that even sublethal levels of SMX can have a synergistic effect with TMP. In the case of the remaining four bacterial strains, the effect of SMX and TMP was additive, i.e., the PTU value was equal to the sum of the toxic effects of the mixture components (Figure 5).

Irradiation in a sunlight simulator resulted in different effects on the toxicity of SMX and TMP mixtures. In the case of *B. diminuta*, sensitive only to SMX, a decrease in SMX concentration resulted in a proportional decrease in toxicity, i.e., MTU and PTU values were similar to each other (Figure 5a). In the case of strains sensitive to both antibiotics to a similar extent (*C. testosteroni* and *P. aurantiaca*), an increase in the MTU to PTU ratio can be observed with increasing irradiation time (Figure 4c and Figure 5d). This may be due to two reasons. First, the synergistic effect was not dependent on maintaining the 5:1 ratio of SMX to TMP, and even a small level of SMX was sufficient for a strong bacteriostatic effect of the mixture. Second, SMX photodegradation products had an effect similar to that of the parent compound.

### 2.4. Analysis of the Photodegradation Products with UPLC-MS/MS

The LC coupled with the high-resolution MS/MS was used to identify the compounds formed during SMX and TMP photodegradation. The analysis revealed the presence of 11 compounds whose abundance after SMX irradiation was significantly higher than in the initial solution (Table 2). The plausible structures of these compounds are given in Figure 6. These derivatives were observed both in individually irradiated samples and in the SMX and TMP mixture as well as in the Biseptol extract. MS/MS fragmentation indicated that two of these derivatives retained the sulfonamide system: SMX 253 and SMX 271. SMX 271 plausibly is a hydroxy derivative of SMX, and MS/MS data indicated that hydroxylation occurred in the isoxazole ring. SMX 253 is an isomer of SMX with the same molecular formula but with a different retention time of 5.1 min versus 13.0 min (in LC-MS/MS analysis), respectively. The derivatives SMX 189 and SMX 191 are desulfonation products of the SMX molecule. Lower molecular weight derivatives are formed by cleavage of the S-N bond. SMX 173 has been identified as a sulfanilic acid (SUA), while SMX 155 has been identified as sulfanylidenecyclohexadieneimine. The end products of SUA transformation were aniline (SMX 93) and 4-iminocyclohexadienone (SMX 107). The 3-amino-5-methylisoxazole (SMX 98, AMI) undergoes cleavage to SMX 100.

In the case of TMP, LC-MS/MS analysis revealed the presence of only five derivatives, the abundance of which was significantly higher in the irradiated samples compared to the unirradiated samples. TMP 125 is probably an imino pyrimidine derivative, while the remaining photoproducts, containing a significant number of oxygen atoms, are probably products of the oxidation of the trimethoxybenzyl part. However, it should be noted that due to the poor decomposition of TMP in the sunlight simulator, the abundance of these derivatives was low and the MS/MS spectra of these compounds are of low quality (Table 3).

## 3. Discussion

### 3.1. Bacteriostatic Activity of SMX and TMP

SMX and TMP are antibiotics often used in human and veterinary medicine, both individually and in a mixture. Their significant ecological importance has been evidenced by the fact that they have been included in the list of pharmaceuticals whose concentration should be monitored in inland waters in the European Union [12,13]. For these reasons, numerous studies have been conducted on methods of their degradation, including photodegradation and photocatalysis [5,8,17,18,19,20,23,27,28,29,30,31,32]. The efficiency of these processes is most often assessed only by chemical analysis methods. Biotests are rarely used, although they allow for the assessment of the biological activity of all substances, not only the parent compounds but also their photodegradation and biodegradation products. In the present work, we used the unique MARA biotest, which enables simultaneous assessment of the sample toxicity towards 11 strains of microorganisms. The bacterial strains used in the MARA test were characterized by varying sensitivity to the antibiotics tested. TMP at concentrations below 1 mg L^−1^ inhibited the growth of more than half of the strains, whereas SMX in this concentration range inhibited only one strain, but the one that was resistant to TMP. So, the use of only one strain of microorganism would not allow for the simultaneous assessment of changes in the process, e.g., photodegradation of both tested antibiotics. Previous studies of various antibiotic toxicities in MARA showed that the strain most sensitive to SMX (*D. diminuta*) was also the most sensitive to oxytetracycline [33] and sulfadiazine [26]. However, the strain most sensitive to TMP (*D. acidovorans*) was also the most sensitive to first-generation fluoroquinolones and equally sensitive to oxytetracycline and ofloxacin [26]. TMP inhibited the growth of *E. coli* strains 01K1H7 [23] and ATCC 25922 [22] at EC_50_ concentrations of 1.0 and 0.31 mg L^−1^, respectively. Our results showed that the toxicity of TMP for *D. acidovorans* expressed as the MTC value was more than 10 times higher, and for the next five strains of the MARA, it was similar (all MTC values below 1 mg L^−1^). Unlike in the case of TMP, *E. coli* ATCC 25922 was not sensitive to SMX with the EC_50_ = 17.1 mg L^−1^ [22]. Using the S-plate test dedicated to detecting sulfonamides with the *Bacillus pumilus* CN 607 strain, Van der Grinten [34] obtained TMP toxicity similar to our results, EC_50_ = 0.028 mg L^−1^. However, in the case of SMX, this strain was over 10 times more sensitive than *D. diminuta* from the MARA, with an EC_50_ value of 0.052 mg L^−1^.

For the first time in ecotoxicological studies, we confirmed the synergistic bacteriostatic effect of TMP and SMX. In the case of 6 out of 10 bacterial strains, the toxicity of the SMX + TMP mixture was more than twice as high as the value predicted based on the concentrations of these antibiotics in the tested solutions. The synergistic effect has been the basis for the recommendations for the use of co-trimoxazole in many diseases, as well as for the inclusion of this combination in the World Health Organization’s list of essential medicines [35]. It also contributes to the need for combined testing of both antibiotics.

### 3.2. SMX and TMP Photodegradation

During 60 min of exposure in the sunlight simulator, the SMX concentration decreased 10-fold, regardless of whether SMX was exposed individually or in a mixture with TMP. The decomposition of SMX was faster than reported in previous publications. Irradiation of SMX at a concentration of 5 mg L^−1^ in a sunlight simulator caused the decomposition of the compound with t½ = 50 min [16]. Among all the tested sulfonamides, SMX was the slowest to photodegrade in the SunTest CPS+ apparatus, with t½ = 227 min [36]. Trovo et al. [15] achieved 40% degradation of SMX in deionized water within 45 min of irradiation in SunTest CPS+. The degradation process was even slower in the study by Gmurek et al. [14], who found that only 50% of the parent compound had been decomposed after 168 h. Without the presence of a photocatalyst, the SMX concentration decreased by only 15% during 40 min of visible light irradiation (λ > 420 nm) [29]. The rate of the SMX photodegradation process depends on the solution pH and decreases with increasing pH [37]. With the increase in pH from 3.0 to 10.0, the rate constants of SMX decreased almost 10 times [16]. However, this should not be the reason for such significant differences in the degree of SMX degradation because both our and the cited experiments were conducted at pH close to neutral. The most probable cause could be the use of different sunlight simulators and/or filters with different ranges of the emitted radiation. SMX absorbs radiation with a wavelength of up to 310 nm (Figure 1). Light simulators, on the other hand, are equipped with filters that cut off ultraviolet with a wavelength below 290–300 nm. Even a minimal shift of the range towards ultraviolet could significantly increase the amount of absorbed radiation and, consequently, the rate of photochemical changes.

Unlike SMX, TMP underwent only a minor 10% degradation during 1 h of exposure in the sunlight simulator. Ljubas et al. [31] observed that the photodegradation of TMP was initially slow (20% concentration reduction within 100 min) and then accelerated significantly, which may indicate the formation of photoproducts that initiate autocatalytic degradation. Similar photodegradation kinetics were observed by Sirtori et al. [8], who indicated that the ketone derivative trimethoxybenzoylpyrimidine may be responsible for this effect. Radiation in the range of λ > 300 nm did not cause any degradation of TMP, while photosensitized degradation in the presence of humic acids was significantly faster [17]. In our studies, we observed a significant increase in the rate of TMP degradation in the mixture with SMX compared to the photodegradation conducted individually. The possibility of TMP photosensitization by other drugs present in water requires further investigation. TMP degradation was studied using different advanced oxidation processes [17], such as ferrate oxidation [23] and TiO_2_ photocatalysis [31].

A number of studies on SMX photodegradation have determined that degradation occurred through cleavage of the molecule at various positions [14,16,28,29,37]. Due to the biological effects, SMX transformation products can be divided into two groups. The first, with preserved sulfonamide structure, is called derivatives, while the second, breakdown products [7]. Derivatives, due to their structural similarity to p-aminosulfonic acid, may have bacteriostatic activity. In our study, LC-MS/MS analyses revealed the presence of two derivatives, SMX 253 and SMX 271. SMX 253 is probably the SMX structural isomer (ISO) reported previously [15,38,39]. It was detected in all SMX samples irradiated both individually and in mixtures. The isomerization of SMX to a derivative in which the sulfonamide group is linked to the isoxazole at position 2 was discussed in detail in Palm et al. [39]. Unlike SMX, which is present in solution mainly in the amine form, ISO is present mainly in the imine form. This results in a change in the absorption spectrum (Figure 1c) and greater resistance to photodegradation [39]. ISO was considered bioactive [38]; however, its bacteriostatic activity has not yet been determined. The 271 Da derivative was detected in earlier studies of SMX photodecomposition [15,36,39]. According to Palm et al. [39], it is an intermediate formed during the isomerization of SMX to ISO, during which the isoxazole ring breaks. It may also be a product of hydroxylation of the isoxazole [15] or benzene [36] ring. Based on the analysis of the MS/MS spectrum, we are more inclined to the latter possibility since this spectrum contains many fragments related to the unchanged SMX fragment containing the benzene ring (e.g., 92.0049, 108.0441, and 156.0111), and no fragments suggesting hydroxylation at this site.

Sulfanilic acid (SUA), sulfanilamide, aniline, and amino-R substituent arising from cleavage of the molecule were the dominant products of direct photolysis of the sulfonamide drugs [37]. SUA and 3-amino-5-methylisoxazole (AMI) were detected by Gmurek et al. [14] and Trovo et al. [15] by exposing SMX to a sunlight simulator. SMX 155 was reported by Martinez et al. [20] and Wang et al. [40], while SMX 107 by Trovo et al. [15]. In our studies, we confirmed the presence of this breakdown photodegradation products but sulfanilamide.

### 3.3. Environmental Significance

Sulfonamides, including SMX, have been detected in sewage and surface waters in many countries. Due to the frequent co-use of the SMX mixture with TMP, TMP was also detected in the same samples [3,9]. Our studies using the multi-species MARA test showed synergistic activity of both antibiotics, not only in a 5:1 mixture but also in the case when SMX had undergone significant photodegradation. This toxicity could be caused by the synergistic action of TMP in combination with low concentrations of SMX or by the resulting photoproducts. Puhlmann et al. [38] found that SMX photodegradation products were toxic to bacteria in a chronic test. Majewsky et al. [7] showed that hydroxy SMX was similarly active as the parent compound. Further studies should be directed towards the analysis of the potential synergistic bacteriostatic effect of TMP with (i) other sulfonamides and (ii) degradation products of SMX and other sulfonamides. Additionally, due to the fact that SMX degradation proceeds differently in seawater than in ionized water [19], it seems interesting to conduct photodegradation analyses in waters of different compositions. The current study was conducted using Tyrode’s medium with a precisely defined composition, a soft water substitute with pH = 7.4. Moreover, it would be advisable to investigate methods that degrade antibiotics in various environmental conditions and also enable the reduction of the overall toxicity of not only the parent compounds but also their degradation products [41,42].

## 4. Materials and Methods

### 4.1. Chemicals

The SMX and TMP standards (purity > 99%) were obtained from Merck (Darmstadt, Germany). The SMX and TMP stock solutions were prepared by dissolving 10.0 mg of the standards in 10 mL of 50% methanol. The working solutions (20 mg L^−1^ SMX or TMP) were prepared just before conducting the tests by diluting the stock solutions with Tyrode’s medium. Tyrode’s medium is a substitute for soft water with low mineral content. It comprises 125 mg NaCl, 3.1 mg KCl, 3.1 mg CaCl_2_, 1.55 mg MgCl_2_, 15.6 mg NaHCO_3_, and 0.78 mg NaH_2_PO_4_ per liter of deionized water [43]. The pH of the Tyrode’s medium was 7.4 ± 0.2, and it did not change after adding the tested antibiotics and during irradiation. The standard mixture containing 20 mg L^−1^ SMX and 4 mg L^−1^ TMP was prepared by mixing the appropriate volumes of the stock solutions. Biseptol pills were purchased from a pharmacy (Biseptol 120, Adamed Pharma S.A., Czosnów, Poland). According to the manufacturer, each pill contained 100 mg SMX and 20 mg TMP. Three pills were placed in 250 mL of 50% methanol and stirred for 30 min. The precipitate was then centrifuged at 10,000 g for 3 min. The antibiotic content in the solution was measured by HPLC with a photodiode array detector. The solution was diluted to a concentration of SMX of 1 mg mL^−1^. The working solution of the Biseptol extract containing 20 mg L^−1^ SMX and 4 mg L^−1^ TMP was prepared just before conducting the tests by diluting the stock solution with Tyrode’s medium.

### 4.2. Photodegradation Experiment

The irradiation process was performed using the SunTest CPS+ apparatus (Klimatest, Warsaw, Poland) (Appendix A) with a 1500 W xenon lamp that provided full-spectrum light (UV/Vis) similar to the sunlight spectrum (Figure 1a). For the photodegradation experiments, the working solutions—SMX (20 mg L^−1^), TMP (20 mg L^−1^), the combination of SMX-TMP (20 + 4 mg L^−1^), and the Biseptol extract (SMX-TMP 20 + 4 mg L^−1^)—were transferred into 60 mL quartz tubes. The samples were irradiated in a temperature-controlled chamber for 30 and 60 min. The temperature during the experiments was 30–35 °C. The fluence rate was set to 750 W m^−2^, which corresponds to the dose of 2700 kJ m^−2^ h^−1^. The experiment was repeated twice.

### 4.3. Liquid Chromatography with Photodiode Array Detector

The concentration of antibiotics was measured using a Shimadzu (Kyoto, Japan) HPLC instrument equipped with an SPD-M10A photodiode array detector (PDA). In the gradient analysis used, phase A consisted of a 0.05% aqueous solution of trifluoroacetic acid, and phase B was an acetonitrile. The HPLC-grade acetonitrile and the trifluoroacetic acid were purchased from Merck (Darmstadt, Germany), while deionized water was procured by using the Milli-Q^®^ Direct water purification system (Merck, Darmstadt, Germany). The separation process was carried out on a LichroCART 50×4 Purospher STAR RP-18 (3 µm) analytical column (Merck, Darmstadt, Germany). The flow rate was 1.0 mL min^−1^, and the concentration of phase B changed according to the following scheme: 0 min: 15%; 1 min: 20%; 8 min: 90%; 9 min: 90%; and 9.10 min: 15%. Quantitative analysis of SMX and TMP was carried out at 268 and 204 nm, respectively. Standard curves of antibiotic solutions were in the concentration range of 0.5–20 mg L^−1^. The limit of quantitation was 0.5 mg L^−1^.

### 4.4. Liquid Chromatography with Mass Spectrometer Detector

Instrumental analysis was performed using a UHPLC Dionex (Germering, Germany) Ultimate 3000 with a Q-Exactive spectrometer, as described previously [43]. Tentative metabolites were detected using Compound Discoverer Software v.3.3 (Thermo Fisher Scientific, Waltham, MA, USA).

### 4.5. Microbial Assay for Risk Assessment

MARA plates and growth media were purchased from NCIMB Ltd. (Aberdeen, UK). The tests were performed according to the standard operational procedure provided by the manufacturer with changes [43]. To show the relative response of each strain in relation to the most sensitive strain, relative sensitivity values (RS) were calculated using the following formula [26]:RS_i_ = MTC_min_/MTC_i_ × 100(1)
where RS_i_ is a RS value for the “i” strain; MTC_i_ is the MTC value of the “i” strain; and MTC_min_ is the MTC value of the most sensitive strain.

The concentration addition approach was used to calculate the predicted toxicity of the samples. The predicted toxicity unit (PTU) was calculated with the following equation:PTU = C_SMX_/MTC_SMX_ + C_TMP_/MTC_TMP_(2)
where C_SMX_ and C_TMP_ are the concentrations of the SMX and TMP in the sample in mg L^−1^, and MTC_SMX_ and MTC_TMP_ are the microbial toxic concentrations of the SMP and TMP expressed in mg L^−1^.

The measured toxicity unit (MTU) was calculated according to the following equation:MTU_i_ = 100/MTC_i_(3)
where MTC_i_ is the microbics toxic concentration of the mixture expressed in % for each MARA strain. The 100% solution was assumed to be 20 and/or 4 mg L^−1^ for SMX and TMP, respectively.

## 5. Conclusions

For the first time, we confirmed the synergistic bacteriostatic effect of TMP and SMX using a multi-species chronic toxicity assay. In the case of 6 out of 10 bacterial strains from the MARA assay, the toxicity of the SMX + TMP mixture was more than twice as high as the value predicted based on the concentrations of these antibiotics in the tested solutions.

Synergistic effects were also observed in samples exposed to SunTest CPS+, where the SMX concentration decreased by 80% and 90% after 30 and 60 min of irradiation. This indicates a possible synergistic effect of the SMX mixture with TMP in proportions other than the recommended 5:1 or a synergistic effect of TMP with the resulting SMX photoproducts. In the SMX irradiated samples, 11 photoproducts were detected, including two with preserved sulfonamide structure.

When assessing the environmental risk of antibiotics, studies of (i) multiple strains of microorganisms and (ii) synergistic bactericidal and/or bacteriostatic effects of drug mixtures are essential.

## Figures and Tables

**Figure 1 molecules-30-01907-f001:**
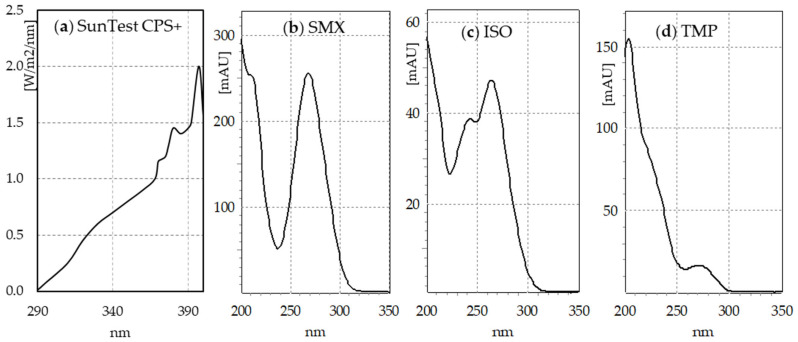
(**a**) Spectrum of simulated light emitted by SunTest CPS+ (based on manufacturer’s data); (**b**) sulfamethoxazole (SMX), (**c**) SMX isomer (ISO), and (**d**) trimethoprim (TMP) absorption spectra.

**Figure 2 molecules-30-01907-f002:**
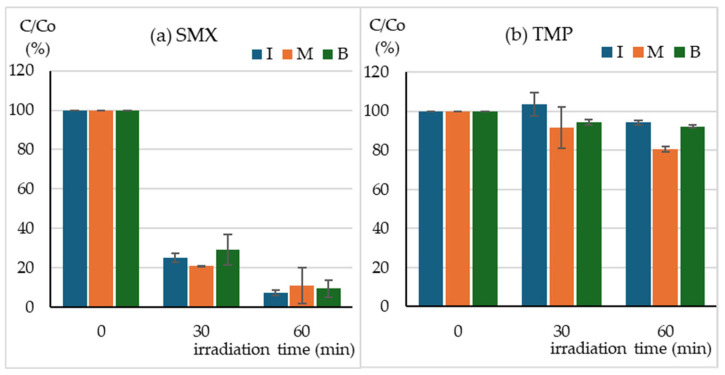
Concentration of (**a**) SMX, (**b**) TMP during irradiation of I—individual solutions, M—mixture of standard solutions, B—Biseptol extract.

**Figure 3 molecules-30-01907-f003:**
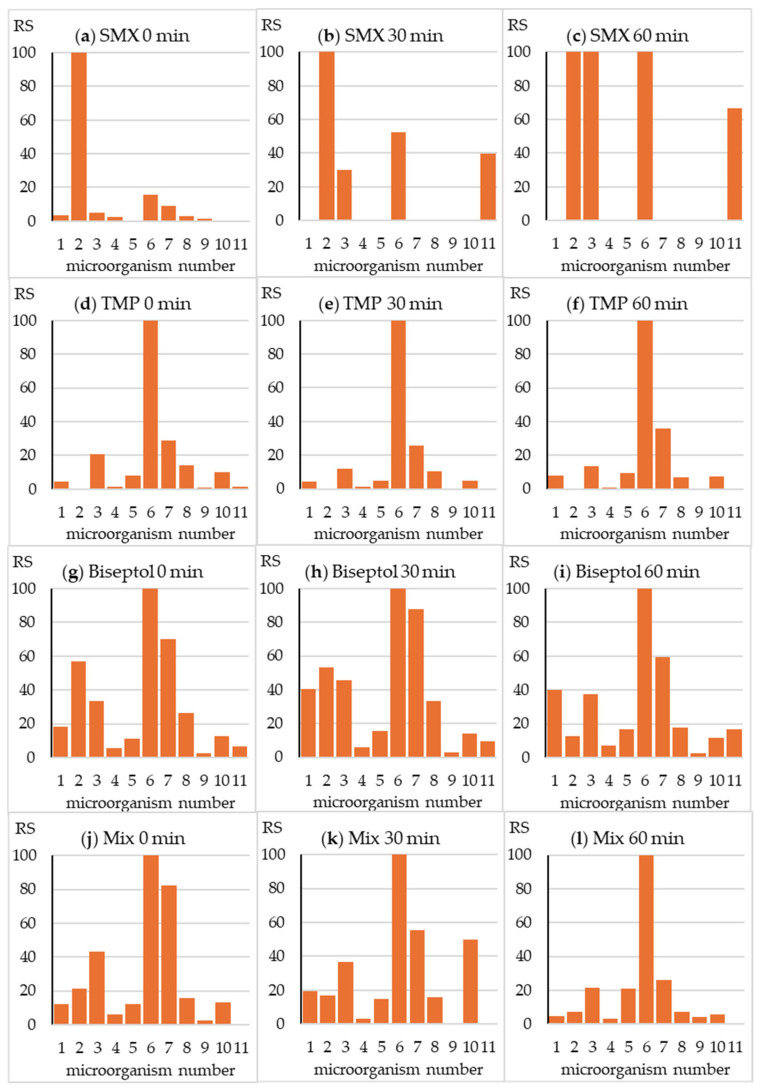
Relative sensitivity (RS) of the microorganisms from the MARA assay. RS_i_ = MTC_min_ /MTC_i_ × 100, where MTC_i_ is the MTC value of the “i” strain and MTC_min_ is the MTC value of the most sensitive strain (No. 6).

**Figure 4 molecules-30-01907-f004:**
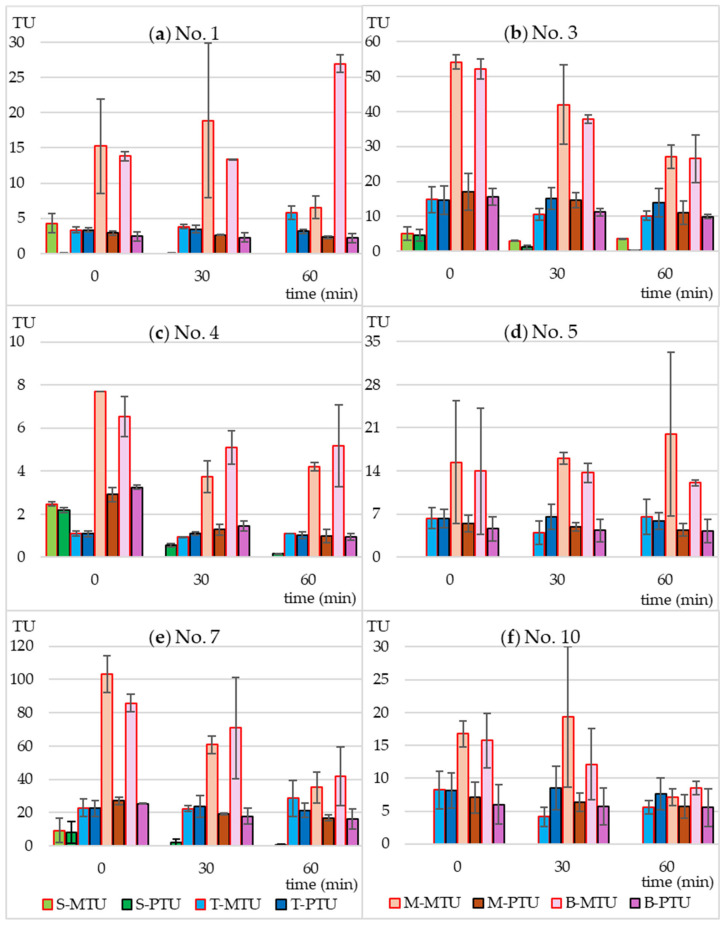
Predicted (PTU) and measured toxicity units (MTU) of the following bacterial strains: (**a**) *Microbacterium* sp.; (**b**) *C. freudii*; (**c**) *C. testosteroni*; (**d**) *E. casseliflavus*; (**e**) *K. gibsonii*; (**f**) *S. rubidaea*. S-MTU and S-PTU—the toxicity values of sulfamethoxazole; T-MTU and T-PTU—the toxicity values of thrimethoprim; M-MTU and M-PTU—the toxicity values of the sulfamethoxazole and thrimethoprim mixture; B-MTU and B-PTU—the toxicity values of the Biseptol extract.

**Figure 5 molecules-30-01907-f005:**
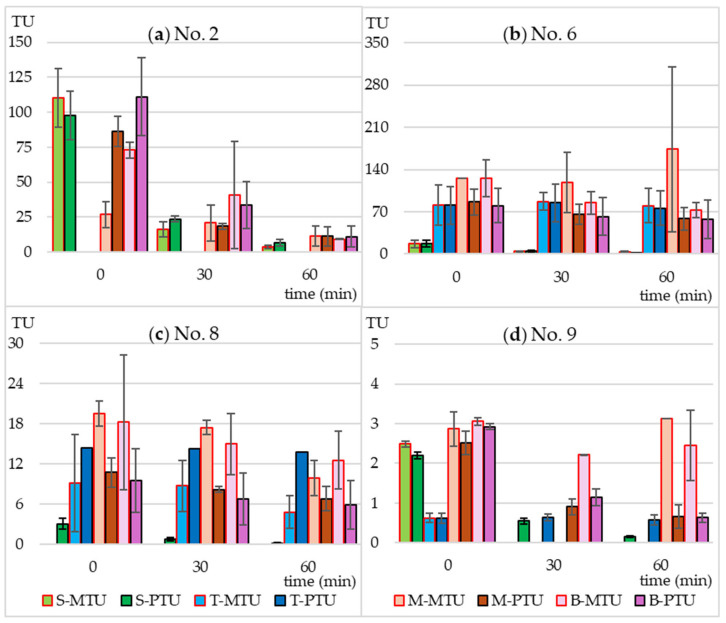
Predicted (PTU) and measured toxicity units (MTU) of the following bacterial strains: (**a**) *B. diminuta*; (**b**) *D. acidovorans*; (**c**) *S. warneri*; (**d**) *P. aurantiaca*. S-MTU and S-PTU—the toxicity values of sulfamethoxazole; T-MTU and T-PTU—the toxicity values of thrimethoprim; M-MTU and M-PTU—the toxicity values of the sulfamethoxazole and thrimethoprim mixture; B-MTU and B-PTU—the toxicity values of the Biseptol extract.

**Figure 6 molecules-30-01907-f006:**
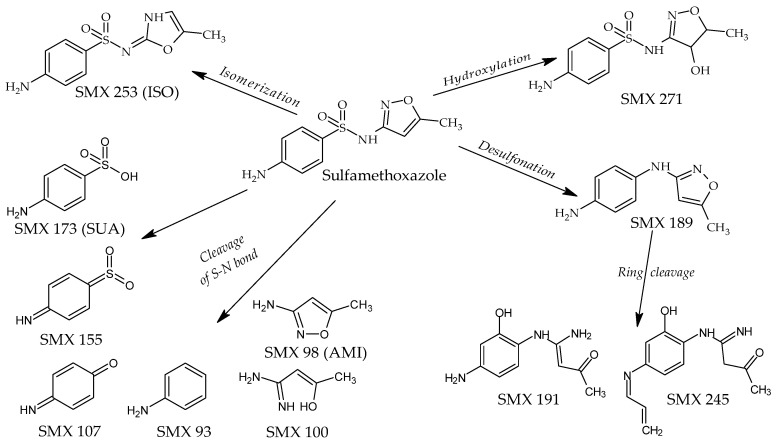
Plausible structures of the SMX photodegradation products. ISO—SMX isomer, SUA—sulfanilic acid, AMI—3-amine-5-methylisoxazole.

**Table 1 molecules-30-01907-t001:** The toxicity of the SMX and TMP in MARA assay. The average ± standard deviation microbics toxic concentration (MTC) values expressed in mg L^−1^.

Microplate Column	Species	Phylogenetic Group	SMX	TMP
1	*Microbacterium* sp.	Gram+	NT ^1^	1.20 ± 0.14
2	*Brevundimonas diminuta*	Gram− α-proteobacteria	0.17 ± 0.01	NT ^1^
3	*Citrobacter freudii*	Gram− γ-proteobacteria	4.07 ± 1.01	0.28 ± 0.07
4	*Comamonas testosteroni*	Gram− α-proteobacteria	7.83 ± 0.50	3.70 ± 0.42
5	*Enterococcus casseliflavus*	Gram+	NT ^1^	0.66 ± 0.18
6	*Delftia acidovorans*	Gram− β-proteobacteria	1.34 ± 0.54	0.054 ± 0.022
7	*Kurthia gibsonii*	Gram+	2.90 ± 2.10	0.18 ± 0.04
8	*Staphylococcus warneri*	Gram+	NT ^1^	0.64 ± 0.51
9	*Pseudomonas aurantiaca*	Gram− γ-proteobacteria	8.05 ± 0.21	6.55 ± 1.20
10	*Serratia rubidaea*	Gram− γ-proteobacteria	NT ^1^	0.52 ± 0.18
11	*Pichia anomala*	Yeast	NT ^1^	4.45 ± 2.76

^1^ Not toxic up to the highest tested concentration of 20.0 mg L^−1^.

**Table 2 molecules-30-01907-t002:** Tentative identification of degradation products of SMX and their MS parameters. Identification was performed based on *m*/*z*, isotopic pattern, and fragmentation spectra (coverage by in silico fragmentation).

Name	Formula	Calc. MW ^1^	Δ Mass (ppm)	RDBE ^2^	H/C ^3^	SFit (%)	Selected MS/MS Fragments (*m*/*z*)
SMX 93	C_6_H_7_N	93.0578	0.07	4	1.2	66	51.0232; 55.0546; 65.0387; 77.0384
SMX 98	C_4_H_6_N_2_O	98.0480	−0.70	3	1.5	49	54.0340; 72.0443; 82.0286
SMX 100	C_4_H_8_N_2_O	100.0637	1.06	2	2.0	79	53.0391; 56.0499; 57.0452; 58.0529; 59.0495; 59.0607; 60.0447; 72.0682
SMX 107	C_6_H_5_NO	107.0371	−0.85	5	0.8	87	53.0389; 65.0385; 80.0493; 81.0333
SMX 155	C_6_H_5_NO_2_S	155.0041	−1.02	5	0.8	77	51.0233; 53.0390; 55.0182; 65.0387; 68.0495; 80.0494; 81.0334; 92.0494; 96.0442; 108.0443; 140.0163
SMX 173	C_6_H_7_NO_3_S	173.0147	0.47	4	1.2	81	65.0387; 81.0338; 92.0496; 108.0443; 156.0114
SMX 189	C_10_H_11_N_3_O	189.0902	−0.06	7	1.1	83	53.0389; 65.0385; 80.0493; 81.0333; 108.0442
SMX 191	C_10_H_13_N_3_O	191.1059	0.05	6	1.3	49	92.0494; 109.0760; 133.0760; 148.0869; 175.0866; 190.0974; 192.1131
SMX 245	C_13_H_15_N_3_O_2_	245.1164	0.03	8	1.2	90	93.0572; 107.0602; 108.0683; 109.0765; 187.0868; 203.1053; 228.1130; 229.0968
SMX 253	C_10_H_11_N_3_O_3_S	253.0521	−0.70	7	1.1	48	68.0494; 92.0492; 108.0441; 156.0110
SMX 271	C_10_H_13_N_3_O_4_S	271.0627	−2.010	6	1.3	53	65.0386; 68.0495; 99.0551; 108.0441; 156.0111; 254.0589

^1^ Monoisotopic molecular mass; ^2^ rings and double bonds equivalent; ^3^ hydrogen versus carbon atoms ratio.

**Table 3 molecules-30-01907-t003:** Tentative identification of degradation products of TMP and their MS parameters. Identification was performed based on *m*/*z*, isotopic pattern, and fragmentation spectra (coverage by in silico fragmentation).

Name	Formula	Calc. MW ^1^	Δ Mass (ppm)	RDBE ^2^	H/C ^3^	SFit (%)	Selected MS/MS Fragments (*m*/*z*)
TMP 125	C_6_H_11_N_3_	125.0953	0.03	3	1.8	74	82.0524; 83.0603; 124.0869
TMP 164	C_6_H_12_O_5_	164.0685	0.56	1	2.0	84	55.0546; 95.0490
TMP 182	C_6_H_12_O_5_	182.0790	−0.05	0	2.3	83	69.0335; 85.0282; 111.0440; 129.0547
TMP 335	C_15_H_29_NO_7_	335.1944	−0.20	2	1.9	93	No MS/MS
TMP 459	C_26_H_37_NO_6_	459.2621	−0.15	9	1.4	87	69.0333; 119.0853; 135.0802

^1^ Monoisotopic molecular mass; ^2^ rings and double bonds equivalent; ^3^ hydrogen versus carbon atoms ratio.

## Data Availability

The original contributions presented in this study are included in the article/Appendix A. Further inquiries can be directed to the corresponding authors.

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
