# Peer review of "Individual and Simultaneous Photodegradation of Trimethoprim and Sulfamethoxazole Assessed with the Microbial Assay for Risk Assessment"

_molecules, 2025, doi:10.3390/molecules30091907_

Round 1
Reviewer 1 Report
Comments and Suggestions for Authors
The authors have presented a study on the individual and simultaneous photodegradation of antibiotics (trimethoprim and sulfametoxazole) to evaluate the change in bacteriostatic activity of the mixture under sunlight irradiation. This topic is relevant and valuable, especially nowadays as these specific antibiotics have been detected in river and inland waters. After carefully reading through the script, here are some points for the authors to consider.
Introduction
Novelty and research hypothesis. While the study is intriguing, the novelty compared to previous research should be more clearly articulated. The end of the Introduction section should explicitly highlight what distinguishes this work from earlier studies and state the specific research objectives o being tested.
Methods
Did the authors notice the evaporation of the solution during irradiation? The quartz tubes containing the solution were irradiated at 30-35°C for one hour. If evaporaton occured than the concentration of solution changed.
What was the pH of the solutions in this study? This should be added as information in the experimental section.
Tyrode's medium composition should be specified (the concentration of individual compounds).
Results
Figure 1– y-axis is not defined
Figure 2 – % is missing on y-axis
Figure 3 and Figure 4 – y-axis is not defined
It would be more informative and clear to the readers if the authors put the degradation pathways instead of structures of photodegradation products (Figure 6).
Disussion
Why did the authors set the exposure time for photodegradation at 30 minutes and 60 minutes?
The kinetics of the process could not be followed and compared to sudies by other researchers if it is set like this.
Are there any studies regarding photodegradation of SMX and TMX in different water matrices, like river water? It would be good to add at the end of discussion where authors mention this (line 343).
Conclusion
The conclusion section should be more concise and quantitative.
By addressing these points, the article will be significantly improved, making it more informative and impactful for a wider scientific audience.
Author Response
Comments 1: The authors have presented a study on the individual and simultaneous photodegradation of antibiotics (trimethoprim and sulfametoxazole) to evaluate the change in bacteriostatic activity of the mixture under sunlight irradiation. This topic is relevant and valuable, especially nowadays as these specific antibiotics have been detected in river and inland waters. After carefully reading through the script, here are some points for the authors to consider.
Introduction. Novelty and research hypothesis. While the study is intriguing, the novelty compared to previous research should be more clearly articulated. The end of the Introduction section should explicitly highlight what distinguishes this work from earlier studies and state the specific research objectives o being tested.
Answer 1: Thank you for this comment. We have added the following sentences to the end of the Introduction: “For the first time, a multi-species bacterial test was used in the analysis of photodegradation of antibiotic mixtures, enabling the assessment of the bacteriostatic activity of the entire sample against different strains of microorganisms.”
Comment 2:
Methods. Did the authors notice the evaporation of the solution during irradiation? The quartz tubes containing the solution were irradiated at 30-35°C for one hour. If evaporaton occured than the concentration of solution changed.
Answer 2: Thank you for your comment. The test tubes used in the analysis are tightly screwed with caps. No evaporation of solutions was observed during 1 or even 4 hours of exposure.
Comments 3: What was the pH of the solutions in this study? This should be added as information in the experimental section. Tyrode's medium composition should be specified (the concentration of individual compounds).
Answer 3: Thank you for your comment. We added the following sentences in the 4.1 paragraph: “It comprises: 125 mg NaCl, 3.1 mg KCl, 3.1 mg CaCl2, 1.55 mg MgCl2, 15.6 mg NaHCO3, and 0.78 mg NaH2PO4 per liter of deionized water [43]. The pH of the Tyrode’s medium was 7.4 ±0.2, and it did not change after adding the tested antibiotics and during irradiation.”
Comments 4: Results. Figure 1– y-axis is not defined; Figure 2 – % is missing on y-axis; Figure 3 and Figure 4 – y-axis is not defined.
Answer 4: Corrected
Comment 5: It would be more informative and clear to the readers if the authors put the degradation pathways instead of structures of photodegradation products (Figure 6).
Answer 5: Figure 6 has been converted into a diagram showing degradation pathways.
Comments 6: Disussion. Why did the authors set the exposure time for photodegradation at 30 minutes and 60 minutes? The kinetics of the process could not be followed and compared to studies by other researchers if it is set like this.
Answer 6: Thank you for these comments. We agree with the reviewer that more frequent measurement of drug concentrations, e.g. every 10 minutes, would allow for the calculation of reaction kinetics and its comparison with other studies. However, the aim of our studies was rather to perform biological analyses at time points at which SMX still had a bacteriostatic effect against most sensitive bacteria (30 min) and did not have such an effect (60 min).
Comment 7: Are there any studies regarding photodegradation of SMX and TMX in different water matrices, like river water? It would be good to add at the end of discussion where authors mention this (line 343).
Answer 7: As in the current project we analyzed the photodegradation process under strictly defined laboratory conditions, we do not want to extend the discussion on the possible effects of additional substances, e.g. dissolved organic matter. Our next project is aimed at assessing the photodegradation of drugs in natural waters.
We changed the ending of the discussion: “Additionally, due to the fact that SMX degradation proceeds differently in sea water than in ionized water [19], it seems interesting to conduct photodegradation analyses in waters of different composition. The current study was conducted using Tyrode’s medium with a precisely defined composition, a soft water substitute with pH=7.4. Moreover, it would be advisable to investigate methods that degrade antibiotics in various environmental conditions and also enable reduction of the overall toxicity of not only the parent compounds but also their degradation products [41,42].”
Comment 8: Conclusion. The conclusion section should be more concise and quantitative.
Answer 8: The conclusion section has been corrected.
Comment 9: By addressing these points, the article will be significantly improved, making it more informative and impactful for a wider scientific audience.
Answer 9: Thank you very much for all your comments, I hope that the corrections made have improved the quality of the manuscript.
Reviewer 2 Report
Comments and Suggestions for Authors
Section 4.2 – please provide a diagram of the SunTest CPS+ apparatus; where was the fluence rate measured? Provide a description of the fluence rate detector.
Author Response
Comment 1:
Section 4.2 – please provide a diagram of the SunTest CPS+ apparatus; where was the fluence rate measured? Provide a description of the fluence rate detector.
Answer 1:
The SunTest CPS+ is a standardized device that provides parameters guaranteed by the manufacturer. A diagram showing the exposure chamber is shown in Figure 1S.
Reviewer 3 Report
Comments and Suggestions for Authors
This work aims to assess the change in bacteriostatic activity during irradiation of the mixture of these antibiotics in a sunlight simulator. The finding is interesting. However, the following aspects still need to be improved:
- The four images in Figure 1 can be displayed overlaid on one image.
- Figure 2: the degradation dynamics are generally presented in a line-point graph with at least five points.
- Figure 4 and Figure 5: some error bars are too large, more repeated experiments are further needed.
- More support can be added for the effect of humic acids in Line 297-298: org/10.1016/j.apsusc.2025.162711
- In addition to plausible structures of the SMX photodegradation products in Figure 6, possible degradation pathway should also be presented.
- More examples on SMX photodegradation can be added in Line 303: doi.org/10.1016/j.cej.2024.158929
- A graph concerning proposed degradation mechanism should also be presented.
- The generation of ROS should be monitored.
Author Response
Comment 1: This work aims to assess the change in bacteriostatic activity during irradiation of the mixture of these antibiotics in a sunlight simulator. The finding is interesting. However, the following aspects still need to be improved:
The four images in Figure 1 can be displayed overlaid on one image.
Answer 1: We tried to merge the spectra of the substances into one figure. Unfortunately, the software of our HPLC system does not allow for placing several spectra obtained in different analyses into one figure.
Comment 2: Figure 2: the degradation dynamics are generally presented in a line-point graph with at least five points.
Answer 2: Thank you very much for this comment. However, the primary goal of our research was not to analyze the kinetics of antibiotic degradation, but to analyze the toxicity of samples during photodegradation. We agree with the reviewer that more frequent measurement of drug concentrations, e.g. every 10 minutes, would allow for the calculation of reaction kinetics and its comparison with other studies. We decided to perform biological analyses at time points at which SMX still had a bacteriostatic effect against most sensitive bacteria (30 min) and did not have such an effect (60 min). And only at these time points we performed chemical analyses.
Comment 3: Figure 4 and Figure 5: some error bars are too large, more repeated experiments are further needed.
Answer 3: The large error bars are due, among others, to the need to use a dilution ratio of 3 in the MARA test, which was associated with large differences in sensitivity between individual bacterial strains. We agree with the reviewer that increasing the number of test repetitions could reduce the error bars. Unfortunately, due to the project being finished, we are currently unable to do additional repetitions. We will take this into account in our next project.
Comment 4: More support can be added for the effect of humic acids in Line 297-298: org/10.1016/j.apsusc.2025.162711
Answer 4: As in the current project we analyzed the photodegradation process under strictly defined laboratory conditions, we do not want to extend the discussion on the possible effects of additional substances, e.g. dissolved organic matter. Our next project is aimed at assessing the photodegradation of drugs in natural waters.
Thank you for pointing out valuable publication. After reading it carefully, we decided that it indicated the direction of future research very well and we cited it at the end of chapter 3.3. “Moreover, it would be advisable to investigate methods that degrade antibiotics in various environmental conditions and also enable reduction of the overall toxicity of not only the parent compounds but also their degradation products [41,42]”.
Comment 5: In addition to plausible structures of the SMX photodegradation products in Figure 6, possible degradation pathway should also be presented.
Answer 5: Figure 6 has been converted into a diagram showing degradation pathways.
Comment 6: More examples on SMX photodegradation can be added in Line 303: doi.org/10.1016/j.cej.2024.158929
Answer 6: We added more examples on SMX photodegradation: “A number of studies on SMX photodegradation have determined that degradation occurred through cleavage of the molecule at various positions [14,16,28,29,37].”
Thank you for pointing out valuable publication. After reading it carefully, we decided that they indicated the direction of future research very well and we cited them at the end of chapter 3.3. “Moreover, it would be advisable to investigate methods that degrade antibiotics in various environmental conditions and also enable reduction of the overall toxicity of not only the parent compounds but also their degradation products [41,42]”.
Comment 7: A graph concerning proposed degradation mechanism should also be presented.
Answer 7: Figure 6 has been converted into a diagram showing degradation pathways.
Comment 8: The generation of ROS should be monitored.
Answer 8: Thank you very much for this valuable comment. In the current project we focused primarily on biological (toxicological) studies. In the next one, aimed at analyzing photodegradation in real (environmental) samples, we will include ROS analyses.
Round 2
Reviewer 1 Report
Comments and Suggestions for Authors
No further comments.
Reviewer 3 Report
Comments and Suggestions for Authors
The authors have made careful revisions and the manuscript can be acceptable now.